# Efficient Forward Architecture Search

**Hanzhang Hu,**[1] **John Langford,**[2] **Rich Caruana,**[2]
**Saurajit Mukherjee,**[2] **Eric Horvitz,**[2] **Debadeepta Dey**[2]
[1]Carnegie Mellon University, [2]Microsoft Research
hanzhang@cs.cmu.edu, {jcl,rcaruana,saurajim,horvitz,dedey}@microsoft.com

## Abstract

We propose a neural architecture search (NAS) algorithm, Petridish, to iteratively
add shortcut connections to existing network layers. The added shortcut connec-
tions effectively perform gradient boosting on the augmented layers. The proposed
algorithm is motivated by the feature selection algorithm forward stage-wise linear
regression, since we consider NAS as a generalization of feature selection for
regression, where NAS selects shortcuts among layers instead of selecting features.
In order to reduce the number of trials of possible connection combinations, we
train jointly all possible connections at each stage of growth while leveraging
feature selection techniques to choose a subset of them. We experimentally show
this process to be an efficient forward architecture search algorithm that can find
competitive models using few GPU days in both the search space of repeatable
network modules (cell-search) and the space of general networks (macro-search).
Petridish is particularly well-suited for warm-starting from existing models crucial
for lifelong-learning scenarios.

## 1 Introduction

Neural networks have achieved state-of-the-art performance on large scale supervised learning tasks
across domains like computer vision, natural language processing, audio and speech-related tasks
using architectures manually designed by skilled practitioners, often via trial and error. Neural
architecture search (NAS) (Zoph & Le, 2017; Zoph et al., 2018; Real et al., 2018; Pham et al.,
2018; Liu et al., 2019; Han Cai, 2019) algorithms attempt to automatically find good architectures
given data-sets. In this work, we view NAS as a bi-level combinatorial optimization problem (Liu
et al., 2019), where we seek both the optimal architecture and its associated optimal parameters.
Interestingly, this formulation generalizes the well-studied problem of feature selection for linear
regression (Tibshirani, 1994; Efron et al., 2004; Das & Kempe, 2011). This observation permits us to
draw and leverage parallels between NAS algorithms and feature selection algorithms.

A plethora of NAS works have leveraged sampling methods including reinforcement learning (Zoph &
Le, 2017; Zoph et al., 2018; Liu et al., 2018), evolutionary algorithms (Real et al., 2017, 2018; Elsken
et al., 2018a), and Bayesian optimization (Kandasamy et al., 2018) to enumerate architectures that
are then independently trained. Interestingly, these approaches are uncommon for feature selection.
Indeed, sample-based NAS often takes hundreds of GPU-days to find good architectures, and can be
barely better than random search (Elsken et al., 2018b).

Another common NAS approach is analogous to sparse optimization (Tibshirani, 1994) or backward
elimination for feature selection, e.g., (Liu et al., 2019; Pham et al., 2018; Han Cai, 2019; Xie et al.,
2019). The approach starts with a super-graph that is the union of all possible architectures, and learns
to down-weight the unnecessary edges gradually via gradient descent or reinforcement learning. Such
approaches drastically cut down the search time of NAS. However, these methods require domain
knowledge to create the initial super-graphs, and typically need to reboot the search if the domain
knowledge is updated.

In this work, we instead take an approach that is analogous to a forward feature selection algorithm and iteratively grow existing networks. Although forward methods such as Orthogonal Matching Pursuit (Pati et al., 1993) and Least-angle Regression (Efron et al., 2004) are common in feature selection and can often result in performance guarantees, there are only a similar NAS approaches (Liu et al., 2017). Such forward algorithms are attractive, when one wants to *expand existing models* as extra computation becomes viable. Forward methods can utilize such extra computational resources without rebooting the training as in backward methods and sparse optimization. Furthermore, forward methods naturally result in a spectrum of models of various complexities to suitably choose from. Crucially, unlike backward approaches, forward methods do not need to specify a finite search space up front making them more general and easier to use when warm-starting from prior available models and for lifelong learning.

Specifically, inspired by early neural network growth work (Fahlman & Lebiere, 1990), we propose a method (Petridish) of growing networks from small to large, where we opportunistically add shortcut connections in a fashion that is analogous to applying gradient boosting (Friedman, 2002) to the intermediate feature layers. To select from the possible shortcut connections, we also exploit sparsity-inducing regularization (Tibshirani, 1994) during the training of the eligible shortcuts.

We experiment with Petridish for both the cell-search (Zoph et al., 2018), where we seek a shortcut connection pattern and repeat it using a manually designed skeleton network to form an architecture, and the less common but more general macro-search, where shortcut connections can be freely formed. Experimental results show Petridish macro-search to be better than previous macro-search NAS approaches on vision tasks, and brings macro-search performance up to par with cell-search counter to beliefs from other NAS works (Zoph & Le, 2017; Pham et al., 2018) that macro-search is inferior to cell-search. Petridish cell-search also finds models that are more cost-efficient than those from (Liu et al., 2019), while using similar training computation. This indicates that forward selection methods for NAS are effective and useful.

We summarize our contribution as follows.

- We propose a forward neural architecture search algorithm that is analogous to gradient boosting on intermediate layers, allowing models to grow in complexity during training and warm-start from existing architectures and weights.
- On CIFAR10 and PTB, the proposed method finds competitive models in few GPU-days with both cell-search and macro-search.
- The ablation studies of the hyper-parameters highlight the importance of starting conditions to algorithm performance.

## 2 Background and Related Work

**Sample-based.** Zoph & Le (2017) leveraged policy gradients (Williams, 1992) to learn to sample networks, and established the now-common framework of sampling networks and evaluating them after a few epochs of training. The policy-gradient sampler has been replaced with evolutionary algorithms (Schaffer et al., 1990; Real et al., 2018; Elsken et al., 2018a), Bayesian optimization (Kandasamy et al., 2018), and Monte Carlo tree search (Negrinho & Gordon, 2017). Multiple search-spaces (Elsken et al., 2018b) are also studied under this framework. Zoph et al. (2018) introduce the idea of cell-search, where we learn a connection pattern, called a cell, and stack cells to form networks. Liu et al. (2018) further learn how to stack cells with hierarchical cells. Cai et al. (2018) evolve networks starting from competitive existing models via net-morphism (Wei et al., 2016).

**Weight-sharing.** The sample-based framework of (Zoph & Le, 2017) spends most of its training computation in evaluating the sampled networks independently, and can cost hundreds of GPU-days during search. This framework is revolutionized by Pham et al. (2018), who share the weights of the possible networks and train all possible networks jointly. Liu et al. (2019) formalize NAS with weight-sharing as a bi-level optimization (Colson et al., 2007), where the architecture and the model parameters are jointly learned. Xie et al. (2019) leverage policy gradient to update the architecture in order to update the whole bi-level optimization with gradient descent.

**Forward NAS.** Forward NAS originates from one of the earliest NAS works by Fahlman & Lebiere (1990) termed "Cascade-Correlation", in which, neurons are added to networks iteratively. Each new neuron takes input from existing neurons, and maximizes the correlation between its activation and

the residual in network prediction. Then the new neuron is frozen and is used to improve the final prediction. This idea of iterative growth has been recently studied in (Cortes et al., 2017; Huang et al., 2018) via gradient boosting (Friedman, 2002). While Petridish is similar to gradient boosting, it is applicable to any layer, instead of only the final layer. Furthermore, Petridish initializes weak learners without freezing or affecting the current model, unlike in gradient boosting, which freezes previous models. Elsken et al. (2018a); Cai et al. (2018) have explored forward search via iterative model changes called net-morphisms (Wei et al., 2016), and control the iterative change via reinforcement learning and evolutionary algorithms. Liu et al. (2017) select models by predicting their performances based on those of some sampled models.

## 3    Preliminaries

**Gradient Boosting:** Let $\mathcal{H}$ be a space of weak learners. Each step of gradient boosting seeks a weak learner $h^* \in \mathcal{H}$ that is the most similar to the negative functional gradient, $-\nabla_{\hat{y}}\mathcal{L}$, of the loss $\mathcal{L}$ with respect to the prediction $\hat{y}$. The similarity is measured by their Frobenius inner product.

$$h^* = \arg\min_{h \in \mathcal{H}} \langle \nabla_{\hat{y}}\mathcal{L}, h \rangle. \tag{1}$$

Then we update the predictor to be $\hat{y} \leftarrow \hat{y} + \eta h^*$, where $\eta$ is the learning rate.

**NAS Optimization:** Given data sample $x$ with label $y$ from a distribution $\mathcal{D}$, a neural network architecture $\alpha$ with parameters $w$ produces a prediction $\hat{y}(x; \alpha, w)$ and suffers a prediction loss $\ell(\hat{y}(x; \alpha, w), y)$. The expected loss is then

$$\mathcal{L}(\alpha, w) = \mathbb{E}_{x,y \sim \mathcal{D}}[\ell(\hat{y}(x; \alpha, w), y)] \approx \frac{1}{|\mathcal{D}_{\text{train}}|} \sum_{(x,y) \in \mathcal{D}_{\text{train}}} \ell(\hat{y}(x; \alpha, w), y). \tag{2}$$

In practice, the loss $\mathcal{L}$ is estimated on the empirical training data $\mathcal{D}_{\text{train}}$. Following (Liu et al., 2019), the problem of neural architecture search can be formulated as a bi-level optimization (Colson et al., 2007) of the network architecture $\alpha$ and the model parameters $w$ under the loss $\mathcal{L}$ as follows.

$$\min_{\alpha} \mathcal{L}(\alpha, w(\alpha)), \quad \text{s.t.} \quad w(\alpha) = \arg\min_{w} \mathcal{L}(\alpha, w) \quad \text{and} \quad c(\alpha) \leq K, \tag{3}$$

where $c(\alpha)$ is the test-time computational cost of the architecture, and $K$ is some constant. Formally, let $x_1, x_2, ...$ be intermediate layers in a feed-forward network. Then a shortcut from layer $x_i$ to $x_j$ ($j > i$) using operation $op$ is represented by $(x_i, x_j, op)$, where the operation $op$ is a unary operation such as 3x3 conv. We merge multiple shortcuts to the same $x_j$ with summation, unless specified otherwise using ablation studies. Hence, the architecture $\alpha$ is a collection of shortcut connections.

**Feature Selection Analogy:** We note that Eq. 3 generalizes feature selection for linear prediction (Tibshirani, 1994; Pati et al., 1993; Das & Kempe, 2011), where $\alpha$ selects feature subsets, $w$ is the prediction coefficient, and the loss is expected square error. Hence, we can understand a NAS algorithm by considering its application to feature selection, as discussed in the introduction and related works. This work draws a parallel to the feature selection algorithm Forward-Stagewise Linear Regression (FSLR) (Efron et al., 2004) with small step sizes, which is an approximation to Least-angle Regression (Efron et al., 2004). In FSLR, we iteratively update with small step sizes the weight of the feature that correlates the most with the prediction residual. Viewing candidate features as weak learners, the residuals become the gradient of the square loss with respect to the linear prediction. Hence, FSLR is also understood as gradient boosting (Friedman, 2002).

**Cell-search vs. Macro-search:** In this work, we consider both cell-search, a popular NAS search space where a network is a predefined sequence of some learned connection patterns (Zoph et al., 2018; Real et al., 2018; Pham et al., 2018; Liu et al., 2019), called cells, and macro-search, a more general NAS where no repeatable patterns are required. For a fair comparison between the two, we set both macro and cell searches to start with the same seed model, which consists of a sequence of simple cells. Both searches also choose from the same set of shortcuts. The only difference is cell-search cells changing uniformly and macro-search cells changing independently.

## 4    Methodology: Efficient Forward Architecture Search (Petridish)

Following gradient boosting strictly would limit the model growth to be only at the prediction layer of the network, $\hat{y}$. Instead, this work seeks to jointly expand the expressiveness of the network at

---

**Algorithm 1** Petridish.initialize_candidates

---

1: **Input**: (1) $L_x$, the list of layers in the current model (macro-search) or current cell (cell-search) in topological order; (2) `is_out(x)`, whether we are to expand at $x$; (3) $\lambda$, hyper parameter for selection shortcut connections.
2: **Output**: (1) $L'_x$, the modified $L_x$ with weak learners $x_c$; (2) $L_c$, the list of $x_c$ created; (3) $\ell_{extra}$, the additional training loss.
3: $L'_x \leftarrow L_x$;  $L_c \leftarrow$ empty list;  $\ell_{extra} \leftarrow 0$
4: **for** $x_k$ in enumerate($L_x$) **do**
5:   **if** not `is_out`($x_k$) **then**  continue  **end if**
6:   Compute the eligible inputs $\text{In}(x_k)$, and index them as $z_1, ..., z_I$.
7:   $x_c \leftarrow \sum_{i=1}^{I} \sum_{j=1}^{J} \alpha_{i,j}^k \text{op}_j(\text{sg}(z_i))$.
8:   Insert the layer $x_c$ right before $x_k$ in $L'_x$.
9:   $\ell_{extra} \leftarrow \ell_{extra} + \lambda \sum_{i=1}^{I} \sum_{j=1}^{J} |\alpha_{i,j}^k|$.
10:   Append $x_c$ to $L_c$.
11:   Modify $x_k$ in $L'_x$ so that $x_k \leftarrow x_k + \text{sf}(x_c)$.
12: **end for**

---

intermediate layers, $x_1, x_2, ....$ Specifically, we consider adding a weak learner $h_k \in \mathcal{H}_k$ at each $x_k$, where $\mathcal{H}_k$ (specified next) is the space of weak learners for layer $x_k$. $h_k$ helps reduce the gradient of the loss $\mathcal{L}$ with respect to $x_k$, $\nabla_{x_k} \mathcal{L} = \mathbb{E}_{x,y \sim \mathcal{D}}[\nabla_{x_k} \ell(\hat{y}(x; \alpha, w), y)]$, i.e., we choose $h_k$ with

$$h_k = \arg\min_{h \in \mathcal{H}_k} \langle h, \nabla_{x_k} \mathcal{L}(\alpha, w) \rangle = \arg\min_{h \in \mathcal{H}_k} \langle h, \mathbb{E}_{x,y \sim \mathcal{D}}[\nabla_{x_k} \ell(\hat{y}(x; \alpha, w), y)] \rangle. \quad (4)$$

Then we expand the model by adding $h_k$ to $x_k$. In other words, we replace each $x_k$ with $x_k + \eta h_k$ in the original network, where $\eta$ is a scalar variable initialized to 0. The modified model then can be trained with backpropagation. We next specify the weak learner space, and how they are learned.

**Weak Learner Space:** The weak learner space $\mathcal{H}_k$ for a layer $x_k$ is formally

$$\mathcal{H}_k = \{\text{op}_{\text{merge}}(\text{op}_1(z_1), ..., \text{op}_{I_{\max}}(z_{I_{\max}})) : z_1, ..., z_{I_{\max}} \in \text{In}(x_k), \text{op}_1, ..., \text{op}_{I_{\max}} \in \text{Op}\}, \quad (5)$$

where $\text{Op}$ is the set of eligible unary operations, $\text{In}(x_k)$ is the set of allowed input layers, $I_{\max}$ is the number of shortcuts to merge together in a weak learner, and $\text{op}_{\text{merge}}$ is a merge operation to combine the shortcuts into a tensor of the same shape as $x_k$. On vision tasks, following (Liu et al., 2019), we set $\text{Op}$ to contain separable conv `3x3` and `5x5`, dilated conv `3x3` and `5x5`, max and average pooling `3x3`, and identity. The separable conv is applied twice as per (Liu et al., 2019). Following (Zoph et al., 2018; Liu et al., 2019), we set $\text{In}(x_k)$ to be layers that are topologically earlier than $x_k$, and are either in the same cell as $x_k$ or the outputs of the previous two cells. We choose $I_{\max} = 3$ through an ablation study from amongst 2, 3 or 4 in Sec. B.5, and we set $\text{op}_{\text{merge}}$ to be a concatenation followed by a projection with conv `1x1` through an ablation study in Sec. B.3 against weighted sum.

**Weak Learning with Weight Sharing:** In gradient boosting, one typically optimizes Eq. 4 by minimizing $\langle h, \nabla_{x_k} \mathcal{L} \rangle$ for multiple $h$, and selecting the best $h$ afterwards. However, as there are $\binom{IJ}{I_{\max}}$ possible weak learners in the space of Eq. 5, where $I = |\text{In}(x_k)|$ and $J = |\text{Op}|$, it may be costly to enumerate all possibilities. Inspired by the parameter sharing works in NAS (Pham et al., 2018; Liu et al., 2019) and model compression in neural networks (Huang et al., 2017a), we propose to jointly train the union of all weak learners, while learning to select the shortcut connections. This process also only costs a constant factor more than training one weak learner. Specifically, we fit the following joint weak learner $x_c$ for a layer $x_k$ in order to minimize $\langle x_c, \nabla_{x_k} \mathcal{L} \rangle$:

$$x_c = \sum_{i=1}^{I} \sum_{j=1}^{J} \alpha_{i,j} \text{op}_j(z_i), \quad (6)$$

where $\text{op}_j \in \text{Op}$ and $z_i \in \text{In}(x_k)$ enumerate all possible operations and inputs, and $\alpha_{i,j} \in \mathbb{R}$ is the weight of the shortcut $\text{op}_j(z_i)$. Each $\text{op}_j(z_i)$ is normalized with batch-normalization to have approximately zero mean and unit variance in expectation, so $\alpha_{i,j}$ reflects the importance of the operation. To select the most important operations, we minimize $\langle x_c, \nabla_{x_k} \mathcal{L} \rangle$ with an L1-

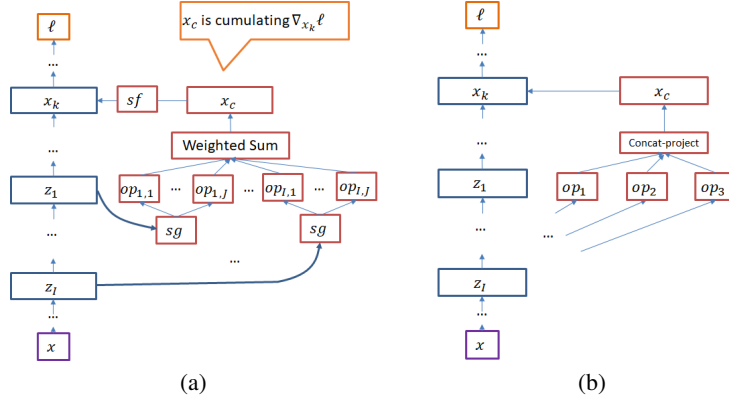

(a)                                                                    (b)

Figure 1: (a) Blue boxes are in the parent model, and red boxes are for weak learning. Operations are joined together in a weighted sum to form $x_c$, in order to match $-\nabla_{x_k}\mathcal{L}$. (b) The top $I_{\max}$ operations are selected and merged with a concatenation, followed by a projection.

---

**Algorithm 2** Petridish.finalize_candidates

---

1: **Inputs**: (1) $L'_x$, the list of layers of the model in topological order; (2) $L_c$, list of selection modules in $L'_x$; (3) $\alpha^k_{i,j}$, the learned operation weights of $x_c$ for layer $x_k$.
2: **Output**: A modified $L'_x$, which is to be trained with backpropagation for a few epochs.
3: **for** $x_c$ in $L_c$ **do**
4:     Let $A = \{\alpha^k_{i,j} : i = 1, ..., I, j = 1, ..., J\}$ be the weights of operations in $x_c$.
5:     Sort $\{|\alpha| : \alpha \in A\}$, and let $\mathtt{op}_1, ..., \mathtt{op}_{I_{\max}}$ be operations with the largest associated $|\alpha|$.
6:     Replace $x_c$ with $\mathtt{proj}(\mathtt{concat}(\mathtt{op}_1, ..., \mathtt{op}_{I_{\max}}))$ in $L'_x$. $\mathtt{proj}$ is to the same shape as $x_k$.
7: **end for**
8: Remove all $\mathtt{sg}(\cdot)$. Replace each $\mathtt{sf}(x)$ with a $\eta x$, where $\eta$ is a scalar variable initialized to 0.

---

regularization on the weight vector $\vec{\alpha}$, i.e.,

$$\lambda\|\vec{\alpha}\|_1 = \lambda \sum_{i=1}^{I} \sum_{j=1}^{J} |\alpha_{i,j}|, \tag{7}$$

where $\lambda$ is a hyper-parameter which we choose in the appendix B.6. $L1$-regularization, known as Lasso (Tibshirani, 1994), induces sparsity in the parameter and is widely used for feature selection.

**Weak Learning Implementation:** A naïve implementation of joint weak learning needs to compute $\nabla_{x_k}\mathcal{L}$ and freeze the existing model during weak learner training. Here we provide a modification to avoid these two costly requirements. Algorithm 1 describes the proposed implementation and Fig. 1a illustrates the weak learning computation graph. We leverage a custom operation called stop-gradient, $\mathtt{sg}$, which has the property that for any $x$, $\mathtt{sg}(x) = x$ and $\nabla_x\mathtt{sg}(x) = 0$. Similarly, we define the complimentary operation stop-forward, $\mathtt{sf}(x) = x - \mathtt{sg}(x)$, i.e., $\mathtt{sf}(x) = 0$ and $\nabla_x\mathtt{sf}(x) = \mathrm{Id}$, the identity function. Specifically, on line 7, we apply $\mathtt{sg}$ to inputs of weak learners, so that $x_c = \sum_{i=1}^{I}\sum_{j=1}^{J} \alpha_{i,j}\mathtt{op}_j(\mathtt{sg}(z_i))$ does not affect the gradient of the existing model. Next, on line 11, we replace the layer $x_k$ with $x_k + \mathtt{sf}(x_c)$, so that the prediction of the model is unaffected by weak learning. Finally, the gradient of the loss with respect to any weak learner parameter $\theta$ is:

$$\nabla_\theta\mathcal{L} = \nabla_{x_k+\mathtt{sf}(x_c)}\mathcal{L}\nabla_{x_c}\mathtt{sf}(x_c)\nabla_\theta x_c = \nabla_{x_k}\mathcal{L}\nabla_\theta x_c = \nabla_\theta\langle\nabla_{x_k}\mathcal{L}, x_c\rangle. \tag{8}$$

This means that $\mathtt{sf}$ and $\mathtt{sg}$ not only prevent the weak learning from affecting the training of existing model, but also enable us to minimize $\langle\nabla_{x_k}\mathcal{L}, x_c\rangle$ via backpropagation on the whole network. Thus, we no longer need explicitly compute $\nabla_{x_k}\mathcal{L}$ nor freeze the existing model weights during weak learning. Furthermore, since weak learners of different layers do not interact during weak learning, we grow the network at all $x_k$ that are ends of cells at the same time.

**Finalize Weak Learners:** In Algorithm 2 and Fig. 1b, we finalize the weak learners. We select in each $x_c$ the top $I_{\max}$ shortcuts according to the absolute value of $\alpha_{i,j}$, and merge them with a

concatenation followed by a projection to the shape of $x_k$. We note that the weighted sum during weak learning is a special case of concatenation-projection, and we use an ablation study in appendix B.3 to validate this replacement. We also note that most NAS works (Zoph et al., 2018; Real et al., 2018; Pham et al., 2018; Liu et al., 2019; Xie et al., 2019; Han Cai, 2019) have similar set-ups of concatenating intermediate layers in cells and projecting the results. We train the finalized models for a few epochs, warm-starting from the parameters in weak learning.

**Remarks:** A key design concept of Petridish is amortization, where we require the computational costs of weak learning and model training to be a constant factor of each other. We further design Petridish to do both at the same time. Following these principles, it only costs a constant factor of additional computation to augment models with Petridish while training the model concurrently.

We also note that since Petridish only grows models, noise in weak learning and model training can result in sub-optimal short-cut selections. To mitigate this potential problem and to reduce the search variance, we utilize multiple parallel workers of Petridish, each of which can warm-start from intermediate models of each other. We defer this implementation detail to the appendix.

## 5 Experiments

We report the search results on CIFAR-10 (Krizhevsky, 2009) and the transfer result on ImageNet (Russakovsky et al., 2015). Ablation studies for choosing the hyper parameters are deferred to appendix B, which also demonstrates the importance of blocking the influence of weak learners to the existing models during weak learning via `sf` and `sg`. We also search on Penn Tree Bank (Marcus et al., 1993), and show that it is not an interesting data-set for evaluating NAS algorithms.

### 5.1 Search Results on CIFAR10

**Set-up:** Following (Zoph et al., 2018; Liu et al., 2019), we search on a shallow and slim networks, which have $N = 3$ normal cells in each of the three feature map resolution, one transition cell between each pair of adjacent resolutions, and $F = 16$ initial filter size. Then we scale up the found model to have $N = 6$ and $F = 32$ for a final training from scratch. During search, we use the last 5000 training images as a validation set. The starting seed model is a modified ResNet (He et al., 2016), where the output of a cell is the sum of the input and the result of applying two 3x3 separable conv to the input. This is one of the simplest seeds in the search space popularized by (Zoph et al., 2018; Pham et al., 2018; Liu et al., 2019). The seed model is trained for 200 epochs, with a batch size of 32 and a learning rate that decays from 0.025 to 0 in cosine decay (Loshchilov & Hutter, 2017). We apply drop-path (Larsson et al., 2017) with probability 0.6 and the standard CIFAR-10 cut-out (DeVries & Taylor, 2017). Weak learner selection and finalization are trained for 80 epochs each, using the same parameters. The final model training is from scratch for 600 epochs on all training images with the same parameters.

**Search Results:** Table 1 depicts the test-errors, model parameters, and search computation of the proposed methods along with many state-of-the-art methods. We mainly compare against models of fewer than 3.5M parameters, since these models can be easily transferred to ILSVRC (Russakovsky et al., 2015) mobile setting via a standard procedure (Zoph et al., 2018). The final training of Petridish models is repeated five times. Petridish cell search finds a model with 2.87±0.13% error rate with 2.5M parameters, in 5 GPU-days using GTX 1080. Increasing filters to $F = 37$, the model has 2.75±0.21% error rate with 3.2M parameters. This is one of the better models among models that have fewer than 3.5M parameters, and is in particular better than DARTS (Liu et al., 2019).

Petridish macro search finds a model that achieves 2.85± 0.12% error rate using 2.2M parameters in the same search computation. This is significantly better than previous macro search results, and showcases that macro search can find cost-effective architectures that are previously only found through cell search. This is important, because the NAS literature has been moving away from macro architecture search, as early works (Zoph et al., 2018; Pham et al., 2018; Real et al., 2018) have shown that cell search results tend to be superior to those from macro search. However, this result may be explained by the superior initial models of cell search: the initial model of Petridish is one of the simplest models that any of the listed cell search methods proposes and evaluates, and it already achieves 4.6% error rate using only 0.4M parameters, a result already on-par or better than any other macro search result.

Table 1: Comparison against state-of-the-art recognition results on CIFAR-10. Results marked with † are not trained with cutout. The first block represents approaches for macro-search. The second block represents approaches for cell-search. We report Petridish results in the format of "best | mean ± standard deviation" among five repetitions of the final training.

| Method | # params (mil.) | Search (GPU-Days) | Test Error (%) |
|---|---|---|---|
| Zoph & Le (2017)† | 7.1 | 1680+ | 4.47 |
| Zoph & Le (2017) + more filters† | 37.4 | 1680+ | 3.65 |
| Real et al. (2017)† | 5.4 | 2500 | 5.4 |
| ENAS macro (Pham et al., 2018)† | 21.3 | 0.32 | 4.23 |
| ENAS macro + more filters† | 38 | 0.32 | 3.87 |
| Lemonade I (Elsken et al., 2018a) | 8.9 | 56 | 3.37 |
| Petridish initial model ($N = 6$, $F = 32$) | 0.4 | – | 4.6 |
| Petridish initial model ($N = 12$, $F = 64$) | 3.1 | – | $3.06 \pm 0.12$ |
| **Petridish macro** | 2.2 | 5 | $2.83 \mid 2.85 \pm 0.12$ |
| NasNet-A (Zoph et al., 2018) | 3.3 | 1800 | 2.65 |
| AmoebaNet-B (Real et al., 2018) | 2.8 | 3150 | $2.55 \pm 0.05$ |
| PNAS (Liu et al., 2017)† | 3.2 | 225 | $3.41 \pm 0.09$ |
| ENAS cell (Pham et al., 2018) | 4.6 | 0.45 | 2.89 |
| Lemonade II (Elsken et al., 2018a) | 3.98 | 56 | 3.50 |
| DARTS (Liu et al., 2019) | 3.4 | 4 | $2.76 \pm 0.09$ |
| SNAS (Xie et al., 2019) | 2.8 | 1.5 | $2.85 \pm 0.02$ |
| Luo et al. (2018)† | 3.3 | 0.4 | 3.53 |
| PARSEC (Casale et al., 2019) | 3.7 | 1 | $2.81 \pm 0.03$ |
| DARTS random (Liu et al., 2019) | 3.1 | – | $3.29 \pm 0.15$ |
| 16 Random Models in Petridish space | $2.27 \pm 0.15$ | – | $3.32 \pm 0.15$ |
| Petridish cell w/o feature selection | $2.50 \pm 0.28$ | – | $3.26 \pm 0.10$ |
| **Petridish cell** | 2.5 | 5 | $2.61 \mid 2.87 \pm 0.13$ |
| **Petridish cell more filters (F=37)** | 3.2 | 5 | $2.51 \mid 2.75 \pm 0.21$ |

Table 2: The performance of the best CIFAR model transferred to ILSVRC. Variance is from multiple training of the same model from scratch. † These searches start from PyramidNet(Han et al., 2017).

| Method | # params (mil.) | # multi-add (mil.) | Search (GPU-Days) | top-1 Test Error (%) |
|---|---|---|---|---|
| Inception-v1 (Szegedy et al., 2015) | 6.6 | 1448 | – | 30.2 |
| MobileNetV2 (Sandler et al., 2018) | 6.9 | 585 | – | 28.0 |
| NASNet-A (Zoph et al., 2017) | 5.3 | 564 | 1800 | 26.0 |
| AmoebaNet-A (Real et al., 2018) | 5.1 | 555 | 3150 | 25.5 |
| PNAS (Liu et al., 2017a) | 5.1 | 588 | 225 | 25.8 |
| DARTS (Liu et al., 2019) | 4.9 | 595 | 4 | 26.9 |
| SNAS (Xie et al., 2019) | 4.3 | 522 | 1.6 | 27.3 |
| Proxyless (Han Cai, 2019)† | 7.1 | 465 | 8.3 | 24.9 |
| Path-level (Cai et al., 2018)† | – | 588 | 8.3 | 25.5 |
| PARSEC (Casale et al., 2019) | 5.6 | – | 1 | 26.0 |
| **Petridish macro (N=6,F=44)** | 4.3 | 511 | 5 | $28.5 \mid 28.7 \pm 0.15$ |
| **Petridish cell (N=6,F=44)** | 4.8 | 598 | 5 | $26.0 \mid 26.3 \pm 0.20$ |

We also run multiple instances of Petridish cell-search, and Table 3 reports performance of the best model of each search run. We observe that the models from the separate runs have similar performances. On average, the search time is 10.5 GPU-days and the model takes 2.8M parameters to achieve 2.88% average mean error rate. In addition, we experiment with replacing feature selection with random choice and leaving all other parts intact, i.e., we keep initialization and finalization of weak learners with parallel workers. The average of mean error rate of the final-trained models is $3.26 \pm 0.04\%$, close to random models, shown near the bottom of Table 1.

**Transfer to ImageNet:** We focus on the mobile setting for the model transfer results on ILSVRC (Russakovsky et al., 2015), which means we limit the number of multi-add per image to be within 600M. We transfer the final models on CIFAR-10 to ILSVRC by adding an initial 3x3

Table 3: Performances of the best models from multiple instances of Petridish cell-search.

| # params (mil.) | Search (GPU-Days) | Test Error (%) |
|---|---|---|
| 3.32 | 7.5 | $2.80 \pm 0.10$ |
| 2.5 | 5 | $2.87 \pm 0.13$ |
| 2.2 | 12 | $2.88 \pm 0.15$ |
| 2.61 | 18 | $2.90 \pm 0.12$ |
| 3.38 | 10 | $2.95 \pm 0.09$ |

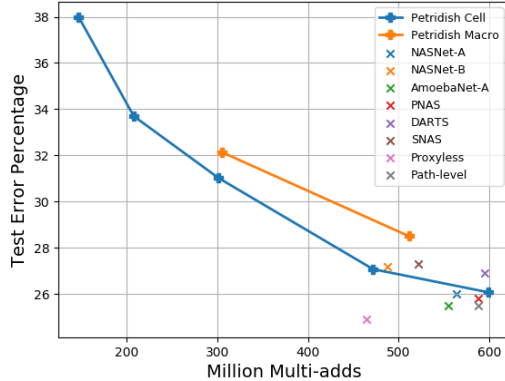

Figure 2: Petridish naturally find a collection of models of different complexity and accuracy. Models outside of the lower convex hull are removed for clarity.

conv of stride of 2, followed by two transition cells, to down-sample the 224x224 input images to 28x28 with $F$ filters. In macro-search, where no transition cells are specifically learned, we again use the the modified ResNet cells from the initial seed model as the replacement. After this initial down-sampling, the architecture is the same as in CIFAR-10 final models. Following (Liu et al., 2019), we train these models for 250 epochs with batch size 128, weight decay $3 * 10^{-5}$, and initial SGD learning rate of 0.1 (decayed by a factor of 0.97 per epoch).

Table 2 depicts performance of the transferred models. The Petridish cell-search model achieves 26.3±0.2% error rate using 4.8M parameters and 598M multiply-adds, which is on par with state-of-the-art results listed in the second block of Table 2. By utilizing feature selection techniques to evaluate multiple model expansions at the same time, Petridish is able to find models faster by one or two orders of magnitude than early methods that train models independently, such as NASNet (Zoph et al., 2018), AmoebaNet (Real et al., 2018), and PNAS (Liu et al., 2017). In comparison to super-graph methods such as DARTS (Liu et al., 2019), Petridish cell-search takes similar search time to find a more accurate model.

The Petridish macro-search model achieves 28.7±0.15% error rate using 4.3M parameters and 511M multiply-adds, a comparable result to the human-designed models in the first block of Table 2. Though this is one of the first successful transfers of macro-search result on CIFAR to ImageNet, the relative performance gap between cell-search and macro-search widens after the transfer. This may be because the default transition cell is not adequate for transfer to more complex data-sets. As Petridish gradually expands existing models, we naturally receive a gallery of models of various computational costs and accuracy. Figure 2 showcases the found models.

## 5.2 Search Results on Penn Treebank

Petridish when used to grow the cell of a recurrent neural network achieves a best test perplexity of $55.85$ and average test perplexity of $56.39 \pm 0.38$ across 8 search runs with different random seeds on PTB. This is competitive with the best search result of (Li & Talwalkar, 2019) of $55.5$ via random search with weight sharing. In spite of good performance we don't put much significance on this particular language-modeling task with this data set because no NAS algorithm appears to perform better than random search (Li & Talwalkar, 2019), as detailed in appendix C.

## 6 Conclusion

We formulate NAS as a bi-level optimization problem, which generalizes feature selection for linear regression. We propose an efficient forward selection algorithm that applies gradient boosting to intermediate layers, and generalizes the feature selection algorithm LARS (Efron et al., 2004). We also speed weak learning via weight sharing, training the union of weak learners and selecting a subet from the union via $L1$-regularization. We demonstrate experimentally that forward model growth can find accurate models in a few GPU-days via cell and macro searches.

## Acknowledgements

We thank J. Andrew Bagnell and Martial Hebert for their support and helpful discussions.

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
