[Supplementary Material]

# A Additional Implementation Details

## A.1 Parallel Workers

Since there are many sources of randomness in model training and weak learning, including SGD batches, drop-path, cut-out, and variable initialization, Petridish can benefit from multiple runs. Furthermore, if one worker finds a cost-efficient model of a medium size, other workers may want the option to warm-start from this checkpoint. Petridish workers warm-start from models on the lower convex hull of the scatter plot of model validation error versus model complexity, because any mixture of other models are either more complex or less accurate.

As there are multiple models on the convex hull, the workers need also choose one at each iteration. To do so, we loop over the models on the hull from the most accurate to the least, and choose a model $m$ with a probability $\frac{1}{n(m)+1}$, where $n(m)$ is the number of times that $m$ is already chosen. This probability is chosen because if a model has been sampled $n$ times, then the next child is the best among the $n+1$ children with probability $\frac{1}{n+1}$. We favor the accurate models, because it is typically more difficult to improve accurate models. In practice, Petridish sample fewer than 100 models, so performances of different sampling algorithms are often indistinguishable, and we settle on this simple algorithm.

## A.2 Select Models for Final Training

The search can be interrupted at anytime, and the best models are the models on the performance convex hull at the time of interruption. For evaluating Petridish on CIFAR-10 (Krizhevsky, 2009), we perform final training on models that are on the search-time convex hull and have near 60 million multi-adds on CIFAR-10 during search with $N = 3$ and $F = 16$. We focus on these models can be translated to the ILSVRC mobile setting easily with a fixed procedure of setting $N = 6$ and $F = 44$.

## A.3 Computation Resources

The search are performed on docker containers that have access to four GPUs. The final training of CIFAR (Krizhevsky, 2009) and PTB (Marcus et al., 1993) models each uses one GPUs. The final training of transferred models on ILSVRC each uses four GPUs. The GPUs can be V100, P100, or GTX1080.

# B Ablation Studies

## B.1 Evaluation Criteria

On CIFAR-10 (Krizhevsky, 2009), we often find that standard deviation of final training and search results to be high in comparison to the difference among different search algorithms. In contrast, the test-error on ILSVRC is more stable, and so that one can more clearly differentiate the performances of models from different search algorithms. Hence, we use ILSVRC transfer results to compare search algorithms whenever the results are available. We use CIFAR-10 final training results to compare search algorithms, if otherwise.

## B.2 Search Space: Direct versus Proxy

This section provides an ablation study on a common theme of recent neural architecture search works, where the search is conducted on a proxy space of small and shallow models, with results transferred to larger models later. In particular, since Petridish uses iterative growth, it need not consider the complexity of a super graph containing all possible models. Thus, Petridish can be applied directly to the final model setting on CIFAR-10, where $N = 6$ and $F = 32$. However, this implies each model takes about eight times the computation, and may introduce extra difficulty in convergence. Table 4 shows the transfer results of the two approaches to ILSVRC. We see that this popular proxy search heuristic indeed leads to more accurate models.

| Method | # params (mil.) | # multi-add (mil.) | Search (GPU-Days) | top-1 Test Error (%) |
|---|---|---|---|---|
| Petridish cell direct (F=40) | 4.4 | 583 | 15.3 | 26.9 |
| **Petridish cell proxy (F=44)** | 4.8 | 598 | 5 | 26.3 |

Table 4: Search space comparison between the direct space of $N = 6$ and $F = 32$ and the proxy space of $N = 3$ and $F = 16$ by evaluating their best mobile setting models on ILSVRC.

Table 5: ILSVRC2012 transfer results. Ablation study on the choice of weighted-sum (WS), concat-projection at the end (CP-end), or the Petridish default merge operation in finalized weak learners. The searches were directly on the search space where $N = 6$ and $F = 32$.

| Method | # params (mil.) | # multi-add (mil.) | Search (GPU-Days) | top-1 Test Error (%) |
|---|---|---|---|---|
| WS macro(F=48) | 5.9 | 756 | 29.5 | 32.5 |
| CP-end macro (F=36) | 5.4 | 680 | 29.5 | 29.1 |
| Petridish macro (F=32) | 4.9 | 593 | 27.2 | 29.4 |
| WS cell (F=48) | 3.3 | 477 | 22.8 | 32.7 |
| CP-end cell (F=44) | 4.7 | 630 | 22.8 | 27.2 |
| **Petridish cell** (F=40) | 4.4 | 583 | 15.3 | 26.9 |

## B.3  $\mathrm{op_{merge}}$: Weighted Sum versus Concatenation-Projection

After selecting the shortcuts in Sec. 4, we concatenate them and project the result with 1x1 conv so that the result can be added to the output layer $x_{out}$. Here we empirically justify this design choice through consideration of two alternatives. We first consider applying the switch only to the final reported model. In other words, instead of using concatenation-projection as the merge operation during search we switch all weak learner weighted-sums to concatenation-projections in the final model, which are trained from scratch to report results. We call this variant CP-end. Another variant where we never switch to concatenation-projection is called WS. Since concatenation-projection incurs additional computation to the model, we increase the channel size of WS variants so that the two variants have similar test-time multiply-adds for fair comparisons. The default Petridish option is switching the weak learner weighted-sums to concatenation-projections each time weak learners are finalized with Alg. 2. We compare WS, CP-end and Petridish on the transfer results on ILSVRC in Table 5, and observe that Petridish achieves similar or better prediction error using less test-time computation and training-time search.

## B.4  Is Weak Learning Necessary?

An interesting consideration is whether to stop the influence of the weak learners to the models during the weak learning. On the one hand, we eventually want to add the weak learners into the model and allow them to be backpropagated together to improve the model accuracy. On the other hand, the introduction of untrained weak learners to trained models may negatively affect training. Furthermore, the models may develop dependency on weak-learner shortcuts that are not selected, which can also negatively affect future models. To study the effects through an ablation study, we remove `sg` and replace `sg` with a variable scalar multiplication that is initialized to zero in Algorithm 1. This is equivalent to adding the joint weak learner $x_c$ of Eq. 6 directly to the boosted layer $x_k$ after random initialization, and then we train the existing model and the joint weak learner together with backpropagation. We call this variant Joint, and compare it against the default Petridish. Table 6 showcases the transfer results of Isolated and Joint to ILSVRC. We compare Petridish cell (F=40) with Joint cell (F=32), two models that have similar computational cost but very different accuracy, and we observe that Isolated leads to much better model than Joint for cell-search. This suggests that the randomly initialized joint weak learners should not directly be added to the existing model to be backpropagated, and the weak learning step is beneficial for the overall search.

## B.5  Number of Merged Operations, $I_{\mathbf{max}}$

As we initialize all possible shortcuts during weak learning, we need decide $I$, the number of them to select for forming the weak learner. On one hand, adding complex weak learners can boost

Table 6: ILSVRC2012 transfer results. Ablation study on the choice of Joint and Isolated for training the weak learners. The search were directly on the search space of $N = 6$ and $F = 32$, different from the proxy space ($N = 3, F = 16$) used in the main text.

| Method | # params (mil.) | # multi-add (mil.) | Search (GPU-Days) | top-1 Test Error (%) |
|---|---|---|---|---|
| Petridish Joint cell (F=32) | 4.0 | 546 | 20.6 | 32.8 |
| **Petridish cell** (F=40) | 4.4 | 583 | 15.3 | 26.9 |

Table 7: Test error rates on CIFAR-10 by models found with different weak learner complexities.

| $I_{\max}$ | Average Lowest Error Rate |
|---|---|
| 2 | 3.08 |
| **3** | 2.88 |
| 4 | 2.93 |

performance rapidly. On the other, this may add sub-optimal weak learners that hinder future growth. We test the choice of $I = 2, 3, 4$ during search. We run with each choice five times, and take the average of their most accurate models that take under 60 million multi-adds on the CIFAR model with $N = 3$ and $F = 16$. Models in this range are chosen, because their transferred models to ILSVRC can have 600 million multi-adds with $N = 6$ and $F = 44$, and hence, they are natural candidate models for ILSVRC mobile setting. Table 7 reports the test error rates on CIFAR10, and we see that $I = 3$ yields the best results.

## B.6 L1 Regularization Constant $\lambda$

We choose the L1 regularization constant $\lambda$ of Eq. 7 to be 0.001 from the range of $\{0.1, 0.001, 0.00001\}$, with the performances of the found models in Table 8. High $\lambda$ means that the $L1$-regularization is highly valued, so that the shortcut selection is more sparse. However, strong regularization also prevents weak learners to fit their target loss gradient well. Since we mainly aim to select the most relevant shortcuts, and not to enforce the strict sparsity, we favor a small regularization constant.

We also note that (Huang et al., 2017a) has previously applied group Lasso to select filters in a DenseNet (Huang et al., 2017b). They apply a changing regularization constant $\lambda$ that gradually increases throughout the training. It will be interesting future improvement to select weak learners through dynamically changed regularization during weak learning.

## C   Search results on Penn Treebank (PTB)

PTB (Marcus et al., 1993) has become a standard dataset in the NAS community for benchmarking NAS algorithms for RNNs. We apply Petridish to search for the cell architectures of a recurrent neural network (RNN) [1]. To keep the results as comparable as possible to most recent and well-performing work we keep the search space the same as used by DARTS (Liu et al., 2019) which in turn is also used byvery recent work (Li & Talwalkar, 2019). There is a set of five primitives {sigmoid, relu, tanh, identity, none} that can be chosen amongst to decide connections between nodes in the cell. We modify the source code provided by Liu et al. (2019) to implement Petridish where we iteratively grow starting from a cell which contains only a single node relu connected to the incoming hidden activation and current input, until we have a total of 9 nodes in the cell to match the size used in DARTS. At each stage of growth we train directly with an embedding size of 850, 25 epochs, 64 batch size and a L1 weight of 10 and select the candidate with the highest L1 weight value. We then add this candidate to the cell by removing the stop-gradient and stop-forward layers and replacing with regular connections. Table 9 shows a summary of the results. The rest of the parameters were kept the same as that used by Liu et al. (2019).

The final genotype obtained from the search procedure is then trained from scratch for 4500 epochs, learning rate of 10 and batch size 64 to obtain final test perplexity reported below. We repeat the

Table 8: Test error rates on CIFAR-10 by models found with different regularization constant $\lambda$.

| Regularization Constant $\lambda$ | Average Lowest Error Rate |
|---|---|
| 0.1 | 3.02 |
| **0.001** | 2.88 |
| 0.00001 | 3.13 |

Table 9: Comparison against state-of-the-art language modeling results on PTB. We report Petridish results in the format of "best | mean $\pm$ standard deviation" from 10 repetitions of the search with different random seeds. * From Table 2 in (Li & Talwalkar, 2019). † (Li & Talwalkar, 2019) report being unable to reproduce the DARTS results and this entry represents the results of DARTS (second order) as obtained via their deterministic implementation. ** (Li & Talwalkar, 2019) report being unable to reproduce ENAS results from original source code. *** ENAS results as reproduced via DARTS source code.

| Method | # params (M) | Search (GPU-Days) | Test Error (perplexity) |
|---|---|---|---|
| Darts (first order) (Liu et al., 2019)* | 23 | 1.5 | 57.6 |
| Darts (second order) (Liu et al., 2019)* | 23 | 2 | 55.7 |
| Darts (second order) (Liu et al., 2019)* † | 23 | 2 | 55.9 |
| ENAS (Pham et al., 2018)** | 24 | 0.5 | 56.3 |
| ENAS (Pham et al., 2018)*** | 24 | 0.5 | 58.6 |
| Random search baseline (Li & Talwalkar, 2019)* | 23 | 2 | 59.4 |
| Random search WS (Li & Talwalkar, 2019)* | 23 | 1.25 | 55.5 |
| **Petridish** | 23 | 1 | 55.85 \| 56.39$\pm$ 0.38 |

search procedure 8 times with different random seeds and report the best and average test perplexity along with the standard deviation across search trials. Table 9 shows the results of running Petridish on PTB. Petridish obtains comparable results to DARTS, ENAS and Random Search WS.

Note that since random search is essentially state-of-the-art search algorithm on PTB[2] we caution the community to not use PTB as a benchmark for comparing search algorithms for RNNs. The merits of any particular algorithm are difficult to compare at least on this particular dataset and task pairing. More research along the lines of Ying et al. (2019) is needed on 1. whether the nature of the search space for RNNs specific to language modeling is particularly amenable to random search and or 2. whether it is the specific nature of RNNs by itself such that random search is competitive on any task which uses RNNs as the hypothesis space. We are presenting the results on PTB for the sake of completion since it has become one of the default benchmarks but ourselves don't derive any particular signal either way in spite of competitive performance.

## Footnotes

[1]Note that for the case of architecture search of RNNs, cell-search and macro-search are equivalent.

[2]As noted by Li & Talwalkar (2019) current human-designed architecture by Yang et al. (2018) still beats the best NAS results albeit using a mixture-of-experts layer which is not in the search space used by DARTS, ENAS, and Petridish to keep results comparable.