[Reviews · NeurIPS 2019]

Reviewer 1



This work includes original ideas and empirical findings. The formulation of weight sharing and using Lasso to find sparse solutions is quite neat and the implementation trick to include weak learners into the existing model is also clear and clean. The study of the effect on the initial model has been ignored in the NAS field and this work brings the importance of it back to the table which we all should pay attention to. The quality of the work is quite good. Related work are carefully reviewed and understood. Many choices in the proposed method are justified either by previous works or the ablation studies in the appendix. The experiments are extensive and informative with round results. The paper is written clearly and I believe this will be an significant work in the NAS field. One minor question for me is the stopping criterion for the experiments: are the experiments run up to 5 days and then report the best? Is it possible to develop some early stopping for NAS?

Reviewer 2



This paper proposes novel neural architecture search method dubbed Petridish which is based on gradient boosting of "weak learners" (i.e. small subnetworks attached to the main network) that are attached to the main network. Originality: The main contribution of the paper is applying basic ideas from gradient-boosting of weak learners to the task of neural architecture search. This is an original idea, which allows a more guided exploration of the space of neural architectures compared to the random steps done, e.g. in evolutionary algorithms. Most related work is adequately discussed. The connection/differences to NAS methods combining network morphisms with evolutionary algorithms should be discussed in more detail as these explore the search space based on similar steps (modifying a model by small incremental additions) but select steps randomly and not based on gradient boosting. Quality: The authors motivate and evaluate the main design decisions of the method carefully. A short summary of the main results from the supplementary material in the main document would be helpful. I am also proposing two control experiments in Point 5 which could further strengthen the paper. Only including models with fewer than 3.5M parameters (which rules out e.g. ProxylessNAS) in Table 1 is somewhat arbitrary. I propose to include at least the models corresponding to the ones in Table 2 (SNAS, ProxylessNAS) for completeness. Clarity: Generally, the paper is very well written and organized. One information missing for being able to reproduce the results (without looking into the code) would be a complete summary of the entire training pipeline of Petridish including data augmentation, regularization etc. Significance: The proposed work is competitive with other recent NAS methods but does not clearly advance the state-of-the-art in terms of search time, test error, number of parameters of the network, or other dimensions. The main significance of the method is in my opinion that it is not restricted to architectures that are subnetworks of a manually defined supergraph. Thus, it allows in principle a more open-ended architecture search without requiring excessive compute resources (since it still allows for weight sharing). This point, however, is only briefly mentioned in the introduction but not explored more thoroughly later on (e.g. in experiments). A more detailed discussion and some experimental evidence whether lifting the requirement of a predefined supergraph is helpful would greatly increase the significance of the paper. Overall, the work introduces an approach for NAS which is novel and presented clearly. The significance of this work is at the moment limited to "yet another NAS approach" (albeit with a nice connection to gradient boosting of weak learners). More clearly carving out the unique advantages of the approach would increase significance. Minor comment: * The bibliography entry for ProxylessNAS uses a wrong order for first and last name of authors.

Reviewer 3



The paper proposes to perform architecture search in the following way. A basic architecture is extended by adding a layer to the side. However during the forward pass that side layer is ignored. During the backward back propagation is used to update the parameters of this layer as if it were contributing during the actual computations. (But gradients are not propagated beyond the layer). Each of the components has an L1 regularised scalar alpha which is also trained that represents their "contribution". These alphas are used in the selection stage to pick the components to add. To me the similarity is that the weak learner in gradient boosting is selected based on the learning process (the gradient) but this computation did not take part in the forward pass. The paper is mostly well written and clear. I am mainly struggling with the iterative process. Are the cells extended once or is this done in an incremental growing manner? This is not described properly in the paper and I would like to see a clarification on this. Could you also clarify this for macro search vs cell search. Are the different layers updated all at once or one by one for macro search? How exactly is the coupling done in cell search? Do you share the same alpha parameters across cells? I think the idea is original, but the evaluation could be improved. Many of the choices were experimentally validated and these results are presented in the appendix. However key experiments are missing. A problem is that the methods it is compared against all use different search spaces. It is unclear whether the benefits come from growing the model/the search space/the actual implementation of the algorithm. For this reason I think the following experiments need to be included 1. (required) Compare the method to the baseline in which you would select a specific model and grow it by randomly selecting operations. This would show that the selection of operation by using the boosting trick is effective. 2. (required) Take the seed model and scale it up/down until it is equally expensive as the final model. This would show that the architectural changes are actually important and that the performance gains do not just come from the additional capacity. Additionally it would be interesting to see whether a more advanced model could be further improved. # Post rebuttal The authors added additional control experiments. Increased the score.

[Author Response · NeurIPS 2019]

We thank all the reviewers for their words of appreciation, suggestions for improving presentation and insightful control experiments. It has made the work better.

**R1** "are the experiments run up to 5 days and then report the best? Is it possible to ... early stopping for NAS?" We keep a search job using four GPUs running until either the best performing models have not changed for a day or the total computation is too much, typically 3-4 days. In the current state, evaluation of a model with final training requires almost as much computation as NAS itself, e.g., CIFAR requires 2 GPU-days final training.

"The parts on multiple workers..." Multiple workers enable the search to reconsider a growth iteration at any of the intermediate models. This reduces the effect of bad early decisions. Furthermore, multiple workers can share intermediate results with each other. Many NAS papers utilize multiple workers like NAS with RL by Zoph et al 2017. Note our reported gpu hours is the sum of all workers' gpu utilization. The ability to use multiple gpus in parallel is very dependent on the search procedure itself. For example DARTS which keeps all possible architectures in a single gpu's memory is hard to parallelize without modifications to the search algorithm itself as in ProxyLessNas.

"..but how the variances for the other methods are not reported" Unfortunately, most other NAS work do not report variance of models or search. We will add the existent ones from AmoebaNet, DARTS, SNAS and PNAS to the paper.

**R2** "The connection/differences to NAS methods combining network morphisms with evolutionary algorithms should be discussed in more detail..." We will add details to summarize search methods based on net-morphism, such as LEMONADE(Elsken et al. 2018) and Path-level(Cai et al. 2018). Both methods also explore the search space with small and iterative incremental changes. However, they choose the increments based on evolutionary algorithms or REINFORCE, where this work aims to guide the changes with gradient information.

"I propose to include at least the models corresponding to the ones in Table 2 (SNAS, ProxylessNAS) for completeness." In general it is difficult to compare NAS algorithms to each other due to differences in search space, size, quality of starting network, search budget used and whether variances are reported to control for stochasticity during training. We have focused on smaller network regimes to keep experimentation manageable and report results there for a fair comparison. However we will change the limit to 4M to include Path-level and ProxyLessNas. Although please note that Path-level and ProxylessNas start with PyramidNet, which is stronger than the similar starting conditions of NASNet, AmoebaNet, DARTS, SNAS, and ENAS.

Comparison to other supergraph methods The main advantage of Petridish compared to other supergraph-based methods is that Petridish doesn't rely on a good supergraph to be made available (which by itself is a manual design decision) and often is not available on datasets which are not cifar10/100/ImageNet on which considerable prior knowledge via manually designed networks exists which informs the supergraph design. Even where supergraphs are available, Petridish can be viewed as breaking the supergraph optimization into multiple steps as opposed to transforming the search into one giant supergraph optimization like DARTS. We are proving out this aspect of Petridish by running on datasets where prior good supergraphs are not available.

Starting from worse models. The initial model of this work is already one of the simplest in the common search space among the NAS works including DARTS, ENAS, NASNet, and AmoebaNet, and we already know from the micro vs. macro comparison that starting condition is a dominating factor for the search result (paper line 229-235).

**R4** "The paper is mostly well written and clear. I am mainly struggling with the iterative process." The cells are incrementally grown for multiple iterations. Each iteration starts with weak-learning with weight sharing (page 4), followed by weak-learner finalization (page 5). Since each weak-learner training with weight sharing does not affect the existing model, we can conduct multiple independent weak-learner trainings simultaneously. In macro search, the layers that are the end of the initial cells grow independently in each iteration at the same time. Cell-search does the same except that we force the same alpha parameters across cells, so that the decision is uniform.

(required) Compare against random growth baseline. **(also requested by R2)** During author response, we constructed 20 models where we grow randomly starting from the initial model. The growth stops when the model computation complexity (in multi-add) is the same or just exceeds the reported model. We train each model 4 times to compute the mean test error rates on CIFAR10. The best mean is 3.03%, and the average mean is $3.32 \pm 0.15\%$, which is close to the random-model performance in DARTS Table 1. In addition, we experiment with replacing feature selection with random choice and leaving all other parts intact, i.e., we keep initialization and finalization of weak learners with parallel workers. The average of mean error rate of the final-trained models is $3.26 \pm 0.04\%$, close to random models.

(required) Compare against enlarged initial models. We created four configurations to enlarge the initial models. The depth are set to 1, 2, 4 and 8 times the depth of the reported models and the number of channels are multiplied by the square root of 8, 4, 2, and 1, so that they have similar complexity as the reported model. These four models have mean error rates ranging from 3.00% to 3.16%, averaged over five instances of final training. In comparison, the mean performance of the reported model of the similar complexity is $2.87 \pm 0.13\%$ error rate.

[Meta-Review · NeurIPS 2019]

This paper introduces a novel method for neural architecture search motivated by the classic idea of gradient boosting for weak learners. The reviewers highlight the elegance of this idea in how it lifts an old idea into an area of contemporary interest. I thus recommend this paper for acceptance.